# Registration and publication of emergency and elective randomised controlled trials in surgery: a cohort study from trial registries

Rachael L Morley,[1,2] Matthew J Edmondson,[1,3] Ceri Rowlands,[1] Jane M Blazeby,[1,3] Robert J Hinchliffe[1,2]

¹Bristol Centre for Surgical Research, Bristol Medical School, University of Bristol, Bristol, UK
²North Bristol NHS Trust, Bristol, UK
³Bristol Royal Infirmary, Bristol, UK

**Correspondence to**
Rachael L Morley;
rm17210@bristol.ac.uk

## ABSTRACT

**Objectives** Emergency surgical practice constitutes 50% of the workload for surgeons, but there is a lack of high quality randomised controlled trials (RCTs) in emergency surgery. This study aims to establish the differences between the registration, completion and publication of emergency and elective surgical trials.

**Design** The clinicaltrials.gov and ISRCTN.com trials registry databases were searched for RCTs between 12 July 2010 and 12 July 2012 using the keyword 'surgery'. Publications were systematically searched for in Pubmed, MEDLINE and EMBASE.

**Participants** Results with no surgical interventions were excluded. The remaining results were manually categorised into 'emergency' or 'elective' and 'surgical' or 'adjunct' by two reviewers.

**Primary outcome measures** Number of RCTs registered in emergency versus elective surgery.

**Secondary outcome measures** Number of RCTs published in emergency versus elective surgery; reasons why trials remain unpublished; funding, sponsorship and impact of published articles; number of adjunct trials registered in emergency and elective surgery.

**Results** 2700 randomised trials were registered. 1173 trials were on a surgical population and of these, 414 trials were studying surgery. Only 9.4% (39/414) of surgical trials were in emergency surgery. The proportion of trials successfully published did not significantly differ between emergency and elective surgery (0.46 vs 0.52; mean difference (MD) −0.06, 95% CI −0.24 to 0.12). Unpublished emergency surgical trials were statistically equally likely to be terminated early compared with elective trials (0.33 vs 0.16; MD −0.18, 95% CI −0.06 to 0.41). Low accrual accounted for a similar majority in both groups (0.43 vs 0.46; MD −0.04, 95% CI −0.48 to 0.41). Unpublished trials in both groups were statistically equally likely to still be planning publication (0.52 vs 0.71; MD −0.18, 95% CI −0.43 to 0.07).

**Conclusion** Fewer RCTs are registered in emergency than elective surgery. Once trials are registered both groups are equally likely to be published.

## INTRODUCTION

It is estimated that up to 50% of the workload of surgical specialties is in emergency

surgical care.[1 2] In low-income and middle-income countries, this figure rises to 60%.[3] Current evidence shows patients undergoing emergency surgery are three times more likely to die than those undergoing elective surgery. For those who survive, they are twice as likely to suffer a complication.[4 5] The UK Emergency Laparotomy Network recently revealed a variation in mortality from 3.6% to 41.7%.[6] Clearly, there is benefit to be gained from more research into emergency surgery to achieve better and more consistent outcomes.[7]

## Strengths and limitations of this study

► Two trial registry databases interrogated giving a large number of registrations.
► Systematic extraction of data from trials registries by two researchers.
► Publication search allowed 63 months from trial registration.
► Difficulty in categorising trials into 'emergency' or 'elective' trials based on registry data, possibility that definitions varied between trials.

Clinical trials are essential for improving patient outcomes. A previous study has found that 52% of surgical randomised controlled trials (RCTs) remain unpublished.[8] There is currently no knowledge about the number of trials registered before publication in emergency surgery. Anecdotally, emergency surgical trials are considered more difficult to organise than their elective counterparts. Barriers to recruitment such as consent, randomisation, ethical and logistical issues can be more challenging in an emergency setting. However, it is not known whether these barriers preclude emergency surgical trials from publication any more often than elective trials.

The primary objective of this study was to determine the number of RCTs registered in emergency versus elective surgery. The secondary objectives were to compare proportions of publication in emergency versus elective surgery; reasons why trials were unpublished; variation by specialty; impact, funding and sponsorship of published trials and the number of adjunct trials published in emergency versus elective surgery.

## METHODS

### Criteria for inclusion and exclusion

Trials registered on two trials databases over a 2-year period between 12 July 2010 and 11 July 2012 that satisfied the following criteria were included: RCTs, at least one arm studying a surgical intervention or intervention within the perioperative period, adult populations and any gender from anywhere in the world. There was no limitation to the comparative group or type of outcome.

Trials with no surgery or with an intervention outside the perioperative period were excluded. Single arm and non-randomised trials were not included as they would unlikely have quality outcome data. Paediatric trials were also excluded as they are likely to have their own difficulties in trial design owing to the nature of involving children.

### Definitions

Surgical interventions were defined as those involving physically changing body tissues and organs through manual operation such as cutting, suturing, abrading or the use of lasers.[9] Within this definition, this study included negative pressure wound therapy but excluded basic wound dressings (eg, self-adhesive dressing). Adjunct trials were defined as any intervention on a population of patients in the perioperative period that would not otherwise be defined as surgery, for example, anaesthesia, post-operative rehabilitation, adjuvant chemotherapy.

Emergency operations were defined as unplanned admissions where it was not possible to discharge the patient home before their operation. This definition was only used when emergency or elective surgery was not specifically mentioned in the inclusion/exclusion criteria and it was not clear from the type of operation studied.

### Search strategy

Two online clinical trials databases were searched, 'ClinicalTrials.gov' and the 'International Standard Randomised Controlled Trial Number (ISRCTN) registry.'[10 11] The registry searches were performed on the same day, 31 October 2016. Search strategies for each registry, respectively, were as follows.

1. Keyword 'surgery' for non-paediatric, phase II–IV, interventional studies registered between 12 July 2010 and 12 July 2012.
2. Keyword 'surgery' for trials registered between 12 July 2010 and 12 July 2012 (paediatric trials were manually excluded).

For RCTs, the average time from registration to completion of data collection and then publication is 24 and 27 months, respectively (51 months in total).[12] The 2010–2012 period was chosen to allow registered trials sufficient time to complete data collection and publication.

Two clinical trial registries were used to capture a range of trials from around the world. This aimed to avoid some selection bias if there was an unknown preference for trials to register with a specific registry.

Expanding the keyword search to 'surgery OR operation' did not return any more results on ClinicalTrials.gov and returned disproportionally more on ISRCTN (4290 compared with 483 for 'surgery' alone). On further scrutiny, searching for 'surgery OR operation' on ISRCTN database appeared to remove all other search parameters and results included studies assigned from 2000 to 2016. It was therefore felt 'surgery' gave the most relevant search results.

The identified RCTs were categorised (surgery vs adjunct, elective vs emergency, or excluded) by two reviewers (RLM and MJE). A pilot was performed where 50 titles were searched by both. Disagreements were discussed and confirmed with a senior researcher (CR and RJH). The full search was then performed with each academic trainee categorising half the trials. If registry data was unclear about the nature of operation, the publications were checked for further information. If this failed to provide adequate information, the corresponding authors were emailed (see email search). Those that could not be categorised and those including both elective and emergency patients within the same study group were excluded from the analysis. RLM and MJE both also extracted data about specialty, sponsorship (industry, hospital, university, government, research institute) and funding (industry, hospital, university, government, research institute, charity or investigator).

### Publication search strategy

The publication search was performed on 19 and 20 October 2017, allowing a minimum 63 months from registration to publication. Trials can manually link their publication to the trials registry database and on occasion they are automatically indexed via PubMed. If this was not the case, then a systematic search using Healthcare Databases Advanced Search (HDAS) was performed (selecting PubMed, Medline and EMBASE). This was repeated in Google to identify any non-indexed publications (eg, conference abstracts). Searches were performed using trial registration number, study title, authors, institutions and keywords. Papers were matched to studies according to trial design, interventions, recruitment numbers, dates of recruitment and hypotheses.[8]

Trials were considered as published if the data appeared in journals or in full or abstract form. Published protocols or experiences from the trials were not counted. Some trials publish their data directly to the trials website and this was counted as a publication if no other publication was found.

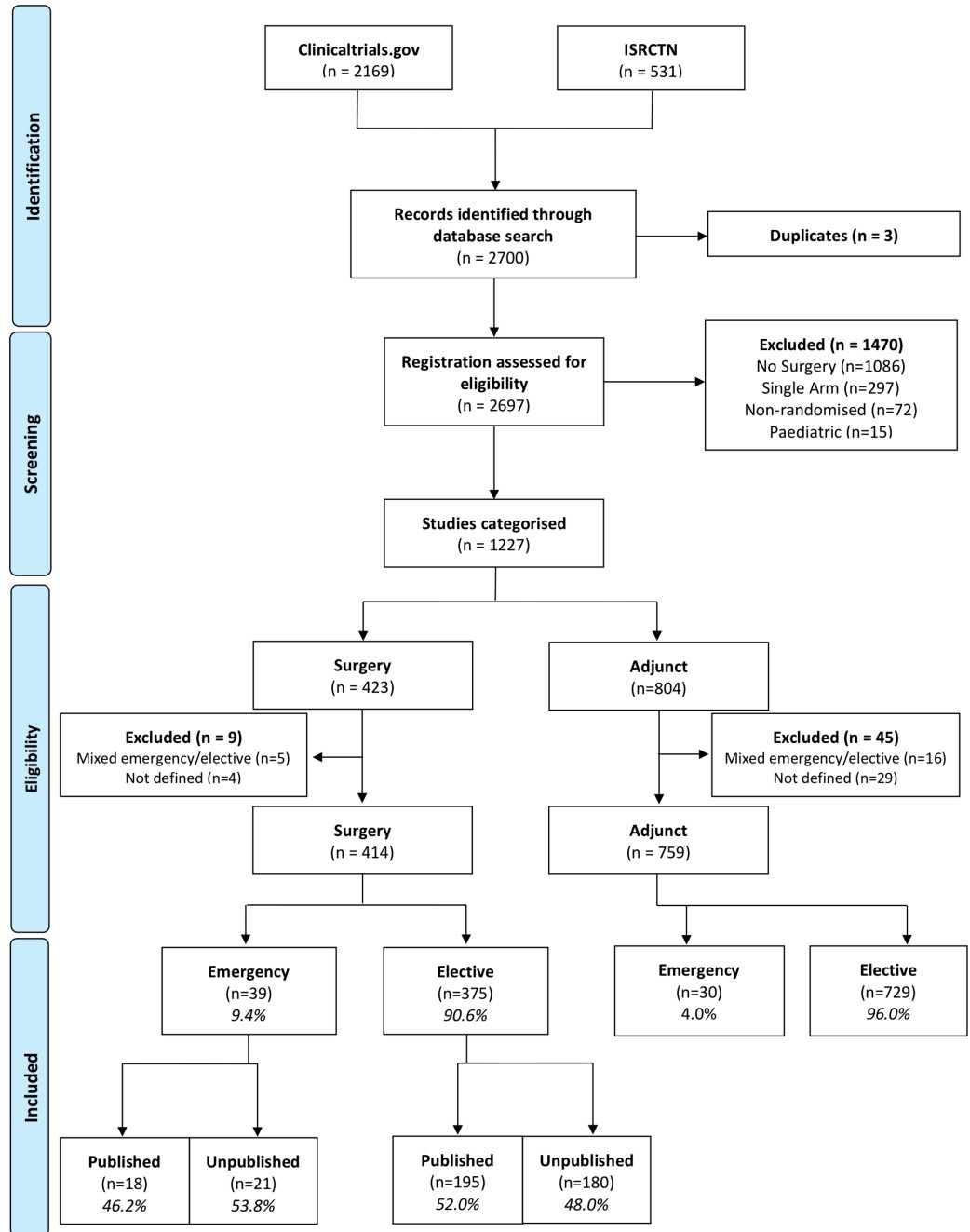

**Figure 1** Preferred Reporting Items for Systematic Reviews and Meta-Analyses (PRISMA) style flow diagram of search results, exclusions and group numbers.

Citation numbers were found by searching for publication title on the Web of Science Core Collection Database to indicate impact of articles. These were used as an indicator of impact of the publications.

### Email search

Emails were identified from those provided on the clinical trials database. If there was no email provided, a Google search for the trial investigator was performed to identify their academic or hospital email. Two standardised emails were sent. The first was sent to seek clarification of inclusion of emergency/elective patients if this remained unclear. The second email was sent to the contacts of all unpublished trials. This clarified publication status and if unpublished, a multiple-choice question of reasons why.

### Statistical analysis

R software (R Core Team 2013. R: A language and environment for statistical computing. Vienna, Austria. ISBN 3-900051-07-0, URL http://www.R-project.org)[13] was used. Differences in means and the test of equal proportions were used to compare data with 95% CIs.[14]

Time to publication was measured from July 2012 until month of publication. Kaplan-Meier graphs were used to visualise the rate of publication in each group.

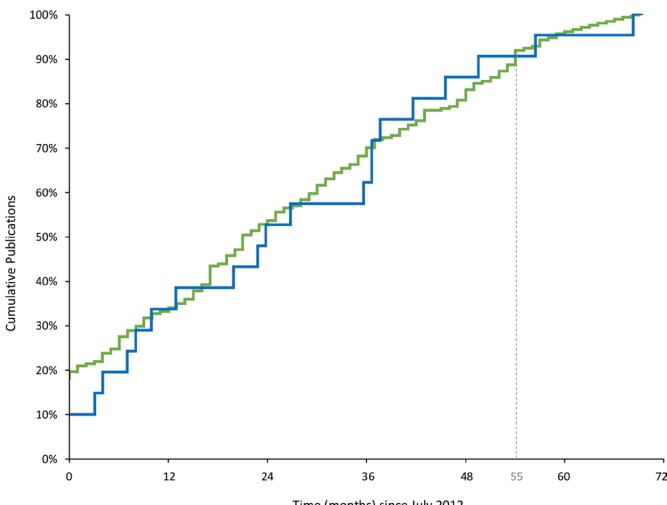

**Figure 2** Kaplan-Meier curves showing publication over time of emergency (blue) and elective (green) surgical trials. Only trials published on or after July 2012 are shown.

## Patient and public involvement

No patients were involved in the design or implementation of this study.

## RESULTS

The initial searches identified 2700 results in total. One thousand four hundred and seventy-three trials were initially excluded (figure 1). Of the remaining 1227 inclusions, 423 were surgical trials and 804 were adjunct trials. Considering only surgical trials, nine were excluded as they could not be defined as emergency or elective (n=4) or specifically included both emergency and elective patients (n=5). This gave a final total of 414 surgical trials in the analysis, of which 39 (9.4%) were emergency and 375 (90.6%) were elective trials.

## Surgical trials

Of the 414 included surgical trials, 213 (51.4%) were published. The publication of emergency surgical trials was not statistically different to elective surgical (0.46 vs 0.52; mean difference (MD) −0.06, 95% CI −0.24 to 0.12). The proportions of publications over time in each group are shown in figure 2, demonstrating a relative plateau at 55 months. There was variation across specialties in the number of emergency and elective surgical trials registered and published (table 1). The number of citations were similar in emergency (mean 21.75, range 0–98) and elective (mean 21.59, range 0–306) surgical trials (difference 0.16, 95% CI −16.7 to 16.9).

Most publications were identified from citations included within trial registrations (n=110). A substantial proportion were identified by searching HDAS and Google (n=90). Few publications were found via email responses (n=4) or data published only on trial registrations (n=9).

In searching for publications and clarifications of definitions, 148 emails were sent. Outcomes from emails

**Table 1** Surgical trials by specialty (excluding specialities with less than 10 trials found) showing proportion of elective versus emergency trials and the percentage of these that were published

| Specialty | Registered; number (% of trials in specialty) | Published; number (% of trials registered) |
| --- | --- | --- |
| General surgery (all) | | |
| Elective | 111 (84.7) | 59 (53.2) |
| Emergency | 20 (15.3) | 13 (65.0) |
| Orthopaedics | | |
| Elective | 86 (86.9) | 38 (44.2) |
| Emergency | 13 (13.1) | 3 (23.1) |
| General (lower gastrointestinal)* | | |
| Elective | 43 (84.3) | 27 (62.8) |
| Emergency | 8 (15.7) | 6 (75.0) |
| General (upper gastrointestinal)* | | |
| Elective | 41 (89.1) | 17 (41.5) |
| Emergency | 5 (10.9) | 3 (60.0) |
| Ophthalmology | | |
| Elective | 34 (100) | 22 (64.7) |
| Emergency | 0 (0) | – |
| Cardiothoracic | | |
| Elective | 33 (97.1) | 18 (54.5) |
| Emergency | 1 (2.9) | 0 (0) |
| Obstetrics and gynaecology | | |
| Elective | 29 (96.7) | 13 (44.8) |
| Emergency | 1 (3.3) | 0 (0) |
| Neurosurgery | | |
| Elective | 24 (92.3) | 8 (33.3) |
| Emergency | 2 (7.7) | 2 (100) |
| Vascular | | |
| Elective | 18 (100) | 11 (61.1) |
| Emergency | 0 (0) | – |
| General (other)* | | |
| Elective | 10 (58.2) | 7 (70.0) |
| Emergency | 7 (41.8) | 4 (57.1) |
| General (breast)* | | |
| Elective | 14 (100) | 6 (42.9) |
| Emergency | 0 (0) | – |
| Urology | | |
| Elective | 14 (100) | 7 (50.0) |
| Emergency | 0 (0) | – |
| Plastics | | |
| Elective | 11 (84.6) | 8 (72.7) |
| Emergency | 2 (15.4) | 0 (0) |
| Other | | |
| Elective | 15 (100) | 11 (73.3) |
| Emergency | 0 (0) | – |
| Total | | |
| Elective | 375 (100) | 195 (52.0) |
| Emergency | 39 (100) | 18 (46.2) |

*Categories combine into general surgery (all). Only general surgery (all) was included in the totals.

**Table 2** Reasons for trials remaining unpublished

|  | Emergency surgery; number (% of total) | Elective surgery; number (% of total) | All, number (%) |
|---|---|---|---|
| Completed and awaiting publication | 7 (33.3) | 62 (34.4) | 69 (34.3) |
| Ongoing data collection | 4 (19.0) | 65 (36.1) | 69 (34.3) |
| Terminated | 7 (33.3) | 29 (15.6) | 36 (17.9) |
| Unknown | 3 (14.3) | 24 (13.9) | 27 (13.4) |
| Total | 21 (100) | 180 (100) | 201 (100) |

Decimals do not add up exactly to total due to rounding.

sent were no response (n=108), undeliverable (n=13) or replied (n=27). Many trials had no contact available (n=69).

Of the 201 unpublished trials, 21 (10.4%) were emergency and 180 (89.6%) were elective surgery. Reasons for non-publication of emergency and elective trials, respectively, were completed but awaiting data analysis/publication process (7/21, 33.3%; 62/180, 34.4%), ongoing data collection (4/21, 19.0%; 65/180, 36.1%), terminated trial (7/21, 33.3%; 28/180, 15.6%) and unknown (3/21, 14.3%; 25/180, 13.9%;).

Unpublished trials in both groups were statistically equally likely to still be planning publication (awaiting data analysis or ongoing data collection; 0.52 vs 0.71; MD −0.18, 95% CI −0.43 to 0.07; table 2). Unpublished emergency surgical trials were statistically equally likely to be terminated early compared with elective trials (0.33 vs 0.16; MD −0.18, 95% CI −0.06 to 0.41). Low accrual accounted for a statistically similar majority in both groups (0.43 vs 0.46; MD −0.04, 95% CI −0.48 to 0.41).

Funding categories were only listed on ISRCTN and not clinicaltrials.gov. Not all trials registered listed sponsorship. Most trials were sponsored by hospitals. Of published trials, there were no significant differences in sponsorship (online Supplementary table 1). Most surgical trials were funded by government bodies (table 3).

### Adjunct trials

Of 804 adjunct trials registered, 45 were excluded as they could not be defined (n=29) or they included both elective and emergency patients (n=16). This left 759 adjunct trials that were analysed. Elective trials represented 96% (n=729), while emergency trials only contributed 4% (n=30). This paper did not consider the non-publication of adjunct trials.

### DISCUSSION

This study found that there are significantly fewer RCTs registered in emergency surgery. However, emergency surgical trials that were registered were equally as likely to be published as elective surgical trials. Recruitment difficulties leading to early trial termination were encountered at a similar rate between elective and emergency studies. Similarly, there was no clear difference between citation numbers in emergency and elective trials, which was used as a marker for research impact. Variation between specialties revealed a comparable pattern to the overall picture.

Registration and publication of all surgical trials has previously been studied.[8] Emergency and elective trials were not studied separately, but found that 48% of all surgical trials remained published at 38 months. The current paper allowed a further 25 months minimum for our publication search and the overall publication proportion was similar (51.5%).

**Table 3** Comparison of the origin of funding between emergency and elective surgical trials. These data was only available from ISRCTN and not clinicaltrials.gov

| Funding | Emergency surgery; number (% of total) | Elective surgery; number (% of total) | All, number (%) |
|---|---|---|---|
| Industry | 0 (0) | 15 (18.1) | 15 (16.5) |
| Hospital | 2 (25.0) | 18 (21.7) | 20 (22.0) |
| University | 0 (0) | 13 (15.7) | 13 (14.3) |
| Government | 5 (62.5) | 18 (21.7) | 23 (25.3) |
| Research institute | 1 (12.5) | 7 (8.4) | 8 (8.8) |
| Charity | 0 (0) | 9 (10.8) | 9 (10.0) |
| Investigator | 0 (0) | 3 (3.6) | 3 (3.3) |
| Total | 8 (100) | 83 (100) | 91 (100) |

Decimals do not add up exactly to total due to rounding.

This suggests a relative plateau in the rate of publication of the trials beyond 38 months. Interpretation of figure 2, however, sees a plateau at around 55 months. While more publications could be expected after this time, extending the search interval may limit applicability to current practice.

The discontinuation of RCTs in critical care has also been studied, which included seven surgical trials.[15] They found the percentage of 'acute care' trials (critical care or care within 24 hours of presentation) was 7%, and a similar publication rate between 'acute' and 'non-acute' trials. However, slow recruitment caused twice as many acute trials to be discontinued than non-acute trials. The methodology was based on comparing published protocols and publications, as opposed to trial registration and publications.

This study has several limitations. Emergency was a binary definition reflecting the wording used within the trials registries. There are inherent difficulties when trying to categorise patients this way, as some operations are 'urgent' but not 'emergency' (eg, bone fractures). It is likely that within the trials searched, the definition of emergency varied. It is possible that this paper gives an overestimation of true emergency surgical trials, where 'urgent' trials have been upgraded to 'emergency' in attempting to define them.

Although this paper demonstrates a lack of registration of emergency surgical trials compared with surgical workload, further work is required to elicit the reasons. Because emergency trials are traditionally perceived as more difficult to achieve, both clinicians and sponsors may be more hesitant to invest their resources. This study shows that emergency surgical trials are equally as likely to be published as elective surgical trials and are a similar investment risk to elective RCTs. Indeed, many issues around consent and randomisation in emergency situations are being overcome.[16–20] With this knowledge, clinicians, researchers and sponsors may feel reassured and inclined to be involved with emergency surgical trials.

Emergency surgery is under-represented in the literature, despite the high volume of workload it creates and poor outcomes. This study shows that registered emergency and elective surgical trials are equally as likely to be published with comparable recruitment issues. A perceived lack of successful completion and reporting of emergency surgical trials should no longer prevent investment into studies that may positively impact emergency surgery.

**Contributors** All authors played a substantial role in the design of the study. RLM synthesised results and wrote the manuscript. RLM and MJE collected data. CR and RJH resolved any conflict in data collection. MJE, CR, JMB and RJH critically revised the manuscript. All authors gave final approval of the version to be published and agree to be held accountable for all aspects of the work.

**Funding** This work was undertaken with the support of the MRC ConDuCT-II (Collaboration and innovation for Difficult and Complex randomised controlled Trials In Invasive procedures) Hub for Trials Methodology Research (MR/K025643/1). This study was supported by the NIHR Biomedical Research Centre at the University Hospitals Bristol NHS Foundation Trust and the University of Bristol. The views expressed are those of the authors and not necessarily those of the UK National Health Service, National Institute for Health Research, or Department of Health.

**Competing interests** None declared.

**Patient consent** Not required.

**Provenance and peer review** Not commissioned; externally peer reviewed.

**Data sharing statement** Data is available upon request from the corresponding author.

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
