## [Reviewer comments · BMJ Open]

ARTICLE DETAILS

TITLE (PROVISIONAL)	REGISTRATION AND PUBLICATION OF EMERGENCY AND ELECTIVE RANDOMISED CONTROLLED TRIALS IN SURGERY: A COHORT STUDY FROM TRIALS REGISTRIES
AUTHORS	Morley, Rachael Lucia; Edmondson, Matthew; Rowlands, Ceri; Blazeby, Jane; Hinchliffe, Robert

VERSION 1 – REVIEW

REVIEWER	Rob JPM Scholten Cochrane Netherlands / Julius Center for Health Sciences and Primary Care, University Medical Center Utrecht, Utrecht University, The Netherlands
REVIEW RETURNED	20-Feb-2018

GENERAL COMMENTS	To me, this is an elegant, easy to follow report about a study that assessed the number of registered + published emergency and elective surgical trials. The methodology is simple and straightforward. I have the following suggestions. General remarks • The section on adjunct surgical trials is very limited and seems to come out of the blue. I am also very interested in the publication numbers of this category (and in the other outcomes). I suggest to either add those to this paper or to indicate more clearly (and in a completely separate section to avoid confusion (see below)) that this has not been studied or will be published in a separate paper. You could also concentrate on the surgical trials only and leave the adjunct surgical trials out. Strengths and limitations • There are many prospective trial registries worldwide (which can be searched simultaneously via the WHO search portal). Searching only two of those (although being the two largest registries) doesn't seem to be a strength. Introduction • Very clear and to the point. Methods • I don't understand why 12th July should produce the most reliable search results. Please, clarify.• Page 6, Line 25-26 "RLM and ME both also extracted data about speciality, sponsorship funding": you mean sponsorship (without 'funding')? If not, what is exactly the difference with 'funding' in the next sentence? Please, clarify.• Page 6, Line 36: 'searching PubMed, MEDLINE and Embase': PubMed is a search engine that searches in MEDLINE. Please,
---

clarify the difference between the two.

- Page 6, Line 45-46: "If it was not found by the above search, data published on the clinical trials website was included as a publication." Not sure, what is meant here. Please, clarify.
- Email search: if no publication could be identified by the Publication search strategy, were trialists then contacted in order to identify publications that could have been missed or that were published in non-indexed journals?

Results

- "Some 1473 trials" (also in the abstract): as a non-native English speaker this gives me the impression of lack of exactness ('we don't really know the exact number, but we think that we found some 1473 trials'). Could you rephrase this, please?
- Page 7, lines 25-26 (and similarly elsewhere in the results section and abstract): "The publication of emergency surgical trials was similar to elective surgical trials (18/39, 46.2% vs 195/375, 52.0%; $p=0.800$). It would be helpful to calculate the difference in proportions / percentages and add a 95% confidence interval thereof (instead of the p-value), like: "18/39, 46.2% vs 195/375, 52.0%; difference -5.8% (95% CI -22.3% to 10.6%)." or "18/39 vs 195/375; difference -0.06 (95% CI -0.22% to 0.11)."
- Page 7, line 28: "The majority of these publications"?
- Page 7, lines 37-49: you might wish to add a Table with these reasons for not (yet) publishing.
- Page 7, lines 51-55: see also general remarks. This part comes out of the blue.
- Page 8: here I'm lost. You appear to elaborate on the adjunct trials from page 7. If this is correct, I would also expect a similar Table (or two extra columns in the Table) for the surgery trials. Or does this section apply to all trials? Please, clarify.
- Page 9 + Table 2+3: I'm lost again. These sections appear to apply to published surgical trials only, but that doesn't seem to be the case. Please, clarify.
- Page 10 Citation numbers: applies to published surgical studies only? Please, clarify and add the numbers of studies. You might also wish to calculate a mean difference of citations + 95% CI.

Discussion

- Clear and concise.
- Page 11, lines 33-38: I'm afraid that I can't follow the reasoning in this paragraph starting with "The timescale of follow up in this study may underestimate the true number of published trials."
- Page 11, line 40 "Although this paper demonstrates a lack of registration of emergency surgical trials": you can't say this if you haven't a gold standard total number of trials. Or do you mean
- "Although this paper demonstrates very few registered emergency surgical trials"?

Tables

- Table 1: please, add a row with the total number of trials (across all specialties) and indicate clearly to what trials these figures apply. To the adjunct trials only or to all trials?
- Tables 2 and 3: please add column %.
- You may wish to add a Table with reasons for not (yet) publishing (based on lines 37-49 on page 7). This could replace the less interesting Table 1, which could be moved to the website?

Figure 1

- Please, include the number of hits in the two databases, then the

	total number of hits and then the number of dual registrations (only 3?). • You may wish to include row percentages in the last two rows (as you do in the text).
--	--

REVIEWER	Bob Siegerink Charite, universitatsmedizin Berlin, Germany
REVIEW RETURNED	05-Mar-2018

GENERAL COMMENTS	This paper, "Registration and Publication of Emergency and Elective Surgical Trials", bmjopen-2018-021700, by Morley et al describes a literature search to see if there are differences between elective and emergency surgeries when it comes to the successful completion and publication of RCT in these areas. The premise is simple: the emergency setting is more complex, so the trials done in that setting are more likely to be successful. The authors provide a good insight to what they did, but some information is still lacking, which i have will describe in a point by point fashion. 1. same data- different analyses for more insight the main analyses presented in the ms is based on simple contingency tables with count data. However, the nature of the data is more "time-to-event" data. I would suggest that adding some analyses that incorporate that element would increase the insights that can be obtained from the data. An example is a kaplan meier curve to show indeed when the plateauing as inferred from literature and own results indeed shows up in this particular set of data. IT is possible that the plateau looks very different for emergence vs non emergency trials. Even when sticking to count data, the authors should consider presenting the associations through ORs and corresponding confidence intervals, as that information provides with more insight how uncertain the results actually are due to the low counts in some subanalyses. 2. lack of description of some data The authors collected a lot of data, including results from the questionnaires that were sent around. The data in the tables only provide limited insight to the wealth of data that they actually have. I propose that the authors rethink the content of the tables and figures to give that data also a good visual representation. Also, some elements seem to be mentioned in the discussion, yet the description of the data in the results section is lacking or only very limited. A rethinking of the way data anad results are presented could remedy this 3. some more insights? some elements that have been proven relevant in these types of studies have not, or only briefly be mentioned in this manuscript. Has any information linked to sample size and power? Where all registration before the start of the trial?
---

VERSION 1 – AUTHOR RESPONSE

Reviewer(s)' Comments to Author:

Reviewer: 1

Reviewer Name: Rob JPM Scholten

Institution and Country: Cochrane Netherlands / Julius Center for Health Sciences and Primary Care, University Medical Center Utrecht, Utrecht University, The Netherlands

Please state any competing interests or state 'None declared': None declared.

Please leave your comments for the authors below

To me, this is an elegant, easy to follow report about a study that assessed the number of registered + published emergency and elective surgical trials. The methodology is simple and straightforward. I have the following suggestions.

General remarks

- The section on adjunct surgical trials is very limited and seems to come out of the blue. I am also very interested in the publication numbers of this category (and in the other outcomes). I suggest to either add those to this paper or to indicate more clearly (and in a completely separate section to avoid confusion (see below)) that this has not been studied or will be published in a separate paper. You could also concentrate on the surgical trials only and leave the adjunct surgical trials out. Thank you, we have moved this paragraph to the end of the results to separate it. We think it still would be useful to have a description as it is included in the figure 1. This is further detailed in the below responses.

Strengths and limitations

- There are many prospective trial registries worldwide (which can be searched simultaneously via the WHO search portal). Searching only two of those (although being the two largest registries) doesn't seem to be a strength.

When we read similar papers, we found that only one trials registry was searched and this is why we included this statement. We have clarified, as the real strength is that we have reviewed a large number of trials.

"Two trials registry databases interrogated giving a large number of registrations."

Introduction

- Very clear and to the point.

Methods

- I don't understand why 12th July should produce the most reliable search results. Please, clarify.

This was due to a glitch in the search function in clinicaltrials.gov. Whatever date we tried to search it defaulted to searching from 12th July. We have left out this sentence to prevent confusion.

- Page 6, Line 25-26 "RLM and ME both also extracted data about speciality, sponsorship funding": you mean sponsorship (without 'funding')? If not, what is exactly the difference with 'funding' in the next sentence? Please, clarify.

Apologies that is a typo and should have read sponsorship and funding. We have amended this.

- Page 6, Line 36: 'searching PubMed, MEDLINE and Embase': PubMed is a search engine that searches in MEDLINE. Please, clarify the difference between the two.

As options is on the Healthcare Databases Advanced Search (HDAS) website it gives the choice to search both and this is what we did. However, as you state we would have covered all in Medline from the Pubmed search. Since this is what we did it might be inaccurate to leave out that we searched Medline.

- Page 6, Line 45-46: "If it was not found by the above search, data published on the clinical trials website was included as a publication." Not sure, what is meant here. Please, clarify.

To clarify, we have changed to, "Some trials publish their data directly to the trials website and this was counted as a publication if no other publication was found."

- Email search: if no publication could be identified by the Publication search strategy, were trialists then contacted in order to identify publications that could have been missed or that were published in non-indexed journals?

Thank you for bringing this to our attention, we had missed this off in error. It now reads, "The second email was sent to all unpublished trials. This clarified publication status and if unpublished, a multiple choice question of reasons why."

Results

- "Some 1473 trials" (also in the abstract): as a non-native English speaker this gives me the impression of lack of exactness ('we don't really know the exact number, but we think that we found some 1473 trials'). Could you rephrase this, please?

We have removed the word 'some.'

- Page 7, lines 25-26 (and similarly elsewhere in the results section and abstract): "The publication of emergency surgical trials was similar to elective surgical trials (18/39, 46.2% vs 195/375,52.0%; $p=0.800$)." It would be helpful to calculate the difference in proportions / percentages and add a 95% confidence interval thereof (instead of the p-value), like: "18/39, 46.2% vs 195/375,52.0%; difference -5.8% (95% CI -22.3% to 10.6%)." or "18/39 vs 195/375; difference -0.06 (95% CI -0.22% to 0.11)."

We have redone the stats with odds ratios for the main part and paired t-test for citations and presented the data in this way with 95% confidence intervals.

- Page 7, line 28: "The majority of these publications"?

Yes that is clearer, we have changed.

- Page 7, lines 37-49: you might wish to add a Table with these reasons for not (yet) publishing. This has been added in

- Page 7, lines 51-55: see also general remarks. This part comes out of the blue.

We have moved this to the end of the results section as it is indeed a separate consideration.

- Page 8: here I'm lost. You appear to elaborate on the adjunct trials from page 7. If this is correct, I would also expect a similar Table (or two extra columns in the Table) for the surgery trials. Or does this section apply to all trials? Please, clarify.

We agree and have moved this sentence up to the publication of emergency/elective surgical trials where it applies to.

- Page 9 + Table 2+3: I'm lost again. These sections appear to apply to published surgical trials only, but that doesn't seem to be the case. Please, clarify.

Again, we have moved this up to fit at the end of the surgical trials results section. Sorry there were some transcription errors into table 2, which we have amended. We have also clarified in the text and table legend that funding data was only available from ISRCTN.

- Page 10 Citation numbers: applies to published surgical studies only? Please, clarify and add the numbers of studies. You might also wish to calculate a mean difference of citations + 95% CI. Again, this has been moved up to fit with the published surgical trials results and 95% confidence intervals stated as above using paired t-test.

Discussion

- Clear and concise.
- Page 11, lines 33-38: I'm afraid that I can't follow the reasoning in this paragraph starting with "The timescale of follow up in this study may underestimate the true number of published trials."

This was intended to explain that if we allowed longer time for publication search it would limit applicability to current practice. It does seem confusing on revisiting, so we have removed this paragraph and changed the paragraph above (beginning "Registration and publication of all surgical trials has previously been studied." to integrate the concept.

- Page 11, line 40 "Although this paper demonstrates a lack of registration of emergency surgical trials": you can't say this if you haven't a gold standard total number of trials. Or do you mean "Although this paper demonstrates very few registered emergency surgical trials"?

We have based this on the idea that 50% of the surgical workload is in emergency surgery and thus one should expect that 50% of the research effort is also into emergency surgery.

We have clarified as "Although this paper demonstrates a lack of registration of emergency surgical trials compared to surgical workload"

Tables

- Table 1: please, add a row with the total number of trials (across all specialties) and indicate clearly to what trials these figures apply. To the adjunct trials only or to all trials?

This has been done and explained more fully in the caption.

- Tables 2 and 3: please add column %.

Have added percentages into existing columns.

- You may wish to add a Table with reasons for not (yet) publishing (based on lines 37-49 on page 7). This could replace the less interesting Table 1, which could be moved to the website?

This has been done.

Figure 1

- Please, include the number of hits in the two databases, then the total number of hits and then the number of dual registrations (only 3?).

This has been done. There are only 3 because generally trials are only registered on one registry as there is no benefit to adding them to another or repeating the same registration on the same registry. This is different to searching publications where a single publication may be indexed on multiple different databases.

- You may wish to include row percentages in the last two rows (as you do in the text).

This has been added

Reviewer: 2

Reviewer Name: bob siegerink

Institution and Country: Charite, universitatsmedizin Berlin, Germany

Please state any competing interests or state 'None declared': none

Please leave your comments for the authors below

This paper, "Registration and Publication of Emergency and Elective Surgical Trials", bmjopen-2018-021700, by Morley et al describes a literature search to see if there are differences between elective

and emergency surgeries when it comes to the successful completion and publication of RCT in these areas. The premise is simple: the emergency setting is more complex, so the trials done in that setting are more likely to be successful. The authors provide a good insight to what they did, but some information is still lacking, which i have will describe in a point by point fashion.

1. same data- different analyses for more insight

the main analyses presented in the ms is based on simple contingency tables with count data. However, the nature of the data is more "time-to-event" data. I would suggest that adding some analyses that incorporate that element would increase the insights that can be obtained from the data. An example is a kaplan meier curve to show indeed when the plateauing as inferred from literature and own results indeed shows up in this particular set of data. IT is possible that the plateau looks very different for emergence vs non emergency trials.

Although we actually had not originally collected data on specific date of publication, we felt that this was a really interesting idea. We have gone through the trials to collect this and create some kaplan meier curves (figure 2) as suggested. We have also compared this in the discussion.

Even when sticking to count data, the authors should consider presenting the associations through ORs and corresponding confidence intervals, as that information provides with more insight how uncertain the results actually are due to the low counts in some subanalyses.

This was mentioned by the above reviewer and has been amended as above.

2. lack of description of some data

The authors collected a lot of data, including results from the questionnaires that were sent around. The data in the tables only provide limited insight to the wealth of data that they actually have. I propose that the authors rethink the content of the tables and figures to give that data also a good visual representation. Also, some elements seem to be mentioned in the discussion, yet the description of the data in the results section is lacking or only very limited. A rethinking of the way data and results are presented could remedy this

We have added a new table to show the reasons for non-publication as a result of data on the trials website and from the email responses.

3. some more insights?

some elements that have been proven relevant in these types of studies have not, or only briefly been mentioned in this manuscript. Has any information linked to sample size and power? Where all registration before the start of the trial?

We took a pragmatic approach to number of trials interrogated, rather than doing a sample size calculation.

Unfortunately we did not gather data on whether trials were registered before or after the trials started. However, from making the Kaplan-Meier graph we can see that 41 were registered before July 2012 and a number were published in July 2012 or very soon after. This probably won't add anything to the paper in this form without doing some more data collection.

VERSION 2 – REVIEW

REVIEWER	Bob Siegerink Center for Stroke research Berlin, Charite, Berlin, Germany
-----------------	--

REVIEW RETURNED	04-Apr-2018
-------------

GENERAL COMMENTS	The authros have taken my comments and adressed them to both in the rebuttal as well as in the manuscript. Some issues remain: // The efforts of extracting extra data is to be applauded: the provided graphs are however currently difficult to interpret. A better approach would be to plot the KM for both emergency and elective in the same graph, with %(and not N) on the y axis. This would allow direct comparison of the two groups. // even though I suggested the ORs as a measure of effect, the other reviewer is right that a difference in % is perhaps the best way to present the data. Even though there is nothing wrong with the OR as such, it is often misinterpreted for a relative risk, and a difference in percentage with confidence interval does not have that difficulty. I sign all my reviews
--

REVIEWER	Rob J.P.M. Scholten Cochrane Netherlands / Julius Center for Health Sciences and Primary Care, University Medical Center Utrecht, The Netherlands
REVIEW RETURNED	10-Apr-2018

GENERAL COMMENTS	The authors should be congratulated with their responses to the comments of the referees and editors. However, I'm afraid that some more challenges have been produced. I also included some comments, that I overlooked in the previous version (for which I also apologise). General remarks 1. The authors have followed the suggestion of the other peer-reviewer with respect to "Even when sticking to count data, the authors should consider presenting the associations through ORs and corresponding confidence intervals, as that information provides with more insight how uncertain the results actually are due to the low counts in some subanalyses." I agree with him, that 95%-CIs provide more insight into the uncertainty that we have. However, there's no need to use ORs here. The OR is a measure of association that is not readily understood. I would make a strong plea for calculating differences in proportions ('risk differences') + 95%-CIs, which enable direct interpretation of the results. You don't even need R for this; you can easily calculate this in a spreadsheet. 2. In addition (and more frightening), the authors appear to have made mistakes with the calculation of the OR. I cite "Of the 414 included surgical trials, 213 (51.4%) were published. The publication of emergency surgical trials was similar to elective surgical trials (46.2% vs 52.0%; Odds ratio 0.51, 95%CI 0.24 - 1.09)". I'm not sure what the authors have done. My reconstruction of the 2*2 table is as follows (see also annex): 18/39 (46.2%) vs 195/375 (52.0%). 'My' OR = 0.79 (95%-CI 0.41 to 1.53) [= (18/21)/(195/180)]. 'My' risk difference (or 'Difference in proportions') would be -0.06 (95%-CI - 0.22 to +0.11) (or, if you wish, -6% (-22% to +11%), which is directly interpretable. Please, check all calculations (or, preferably) replace all ORs with RDs (+95%-CIs). 3. I'm also very confused by the use of the paired t-test (page 7 - lines 6-7: A paired t-test was used to compare citation numbers and odds ratios used all other comparisons at 95% confidence intervals). A paired t-test is used for paired (or dependent) data, e.g. the difference between baseline and follow-up in a series of patients. In
---

this study, we have independent data and, therefore, the independent t-test should be used. This also produces estimates of the difference in means (e.g. mean number of citations) and a 95%-CI. NB: this requires normally distributed data. Let's look at page 7 - lines 30-32: "Citation numbers were similar in emergency (mean 21.75, range 0-98) and elective (mean 21.59, range 0-306) surgical trials (difference 0.16 95%CI -16.7 to 16.9)". I can't replicate this (not sure if this is correct; the 95% CI is extremely wide). If you calculate a difference of means with a 95% CI, one assumes Normal distributions. In that case one should mention the standard deviations (instead of or additional to the ranges that are reported now). However, I expect that these data are severely non-normal, so a non-parametric statistical test is indicated (comparing medians instead of means) and medians and ranges should be reported. One could question, however, whether in this case a test is indicated (in view of the trivial difference of 0.16 citations).

Specific remarks

4. Title: to me, this is not a systematic review. What about a (historical) cohort study (of surgical trials registered in trial registries)?
 5. Page 7 - Table 1: the totals (360-39) don't match with the text (375-39). Add a row with Other (or Remaining or Not classified) to match the totals in the text.
 6. Page 8 - Lines 41-42: "Unpublished emergency surgical trials were equally likely to be terminated early compared with elective trials (33.3% vs 15.6%; OR 2.4, 95%CI 0.88 - 6.40)." This is an incorrect conclusion (and an incorrect OR), which also applies to other instances, e.g. in "Unpublished trials in both groups were equally likely to still be planning publication (52.4% vs 70.1%; OR 0.50, 95%CI 0.20 - 1.25)". Not-significant is not similar to equality or lack of effect. Formally one should say "A difference of having been terminated early between emergency and elective trials could be demonstrated nor refuted."
 7. Page 8-9: Table 2 is not referred to in the text (or I missed it).
 8. Page 9 - Lines 23-24: "Of published trials, there was no significant differences in sponsorship (supplementary table 1)". Please, correct the English.
 9. Discussion: please, check the conclusions (see also my previous remarks about incorrect inferences). This also applies to the abstract.
 10. Figure 2: it's very confusing to have another format for C (which covers the whole width of the page). The three (or two) curves could be presented in one graph. In addition, no methods are mentioned for the Kaplan-Meier curves / time to publication. In the results, there's only one line ("The rate of publications, in each group and overall, are shown in figure 2." without any interpretation (or log-rank test to test the hypothesis that time to publication is equal between the two types of surgery).
- Overlooked in the previous version (apologies):
11. Please, replace "randomised control trials" with "randomised controlled trials" (overlooked in the previous version).
 12. Page Line 41: "Fewer randomised clinical trials": please, be consistent with labelling (->"Fewer randomised controlled trials")
 13. Page 4 Line 25: "The secondary objectives were to compare rates of ...". You don't compare rates (which have an element of time in it: number of events / person time), but proportions. Please, correct.
 14. Page 5: two search strategies are presented. I may miss the point completely, but to me, strategy 2 completely covers strategy 1. Please, explain the difference (and the need for strategy 1).
 15. Page 6, line 11: The searches were categorised -> The identified

	RCTs were categorised 16. Page 6 - Line 52-53: The second email was sent to all unpublished trials -> to the contact persons of unpublished trials 17. Page 7 - Line 28: "The rate of publications, in each group and overall, are shown in". As mentioned before: rates are about incidence densities. Proportion is a better label. English grammar should be corrected ('rate are shown' -> proportionS are shown). 18. Page 8 – Lines 23-25: "The majority of these publications were [was?] cited within trial registrations (n=110)." Do you mean: The majority of publications was found / identified / discovered by citations that were included in the trial registrations? 19. Page 8 – Lines 32-27: "Reasons for non-publication of emergency and elective trials respectively were and unknown". How can a reason be unknown? Didn't the trialists know why they stopped the trial or did they not answer this question (-> 'no reason provided')?
--	---

VERSION 2 – AUTHOR RESPONSE

Reviewer: 2

Reviewer Name: Bob Siegerink

Institution and Country: Center for Stroke research Berlin, Charite, Berlin, Germany Please state any competing interests or state 'None declared': none declared

Please leave your comments for the authors below The authors have taken my comments and adressed them to both in the rebuttal as well as in the manuscript. Some issues remain:

// The efforts of extracting extra data is to be applauded: the provided graphs are however currently difficult to interpret. A better approach would be to plot the KM for both emergency and elective in the same graph, with %(and not N) on the y axis. This would allow direct comparison of the two groups.

We have done this for emergency and elective groups and left out the total (previously C).

// even though I suggested the ORs as a measure of effect, the other reviewer is right that a difference in % is perhaps the best way to present the data. Even though there is nothing wrong with the OR as such, it is often misinterpreted for a relative risk, and a difference in percentage with confidence interval does not have that difficulty.

This has been edited as requested.

I sign all my reviews - Bob Siegerink

Reviewer: 1

Reviewer Name: Rob J.P.M. Scholten

Institution and Country: Cochrane Netherlands / Julius Center for Health Sciences and Primary Care, University Medical Center Utrecht, The Netherlands Please state any competing interests or state 'None declared': None declared

Please leave your comments for the authors below The authors should be congratulated with their responses to the comments of the referees and editors. However, I'm afraid that some more challenges have been produced. I also included some comments, that I overlooked in the previous version (for which I also apologise).

General remarks

1. The authors have followed the suggestion of the other peer-reviewer with respect to "Even when sticking to count data, the authors should consider presenting the associations through ORs and corresponding confidence intervals, as that information provides with more insight how uncertain the results actually are due to the low counts in some subanalyses." I agree with him, that 95%-CIs provide more insight into the uncertainty that we have. However, there's no need to use ORs here. The OR is a measure of association that is not readily

understood. I would make a strong plea for calculating differences in proportions ('risk differences') + 95%-CIs, which enable direct interpretation of the results. You don't even need R for this; you can easily calculate this in a spreadsheet.

We have done as you have suggested changing previously calculated odds ratios to differences of means and associated 95% confidence intervals.

2. In addition (and more frightening), the authors appear to have made mistakes with the calculation of the OR. I cite "Of the 414 included surgical trials, 213 (51.4%) were published. The publication of emergency surgical trials was similar to elective surgical trials (46.2% vs 52.0%; Odds ratio 0.51, 95%CI 0.24 - 1.09)". I'm not sure what the authors have done. My reconstruction of the 2*2 table is as follows (see also annex): 18/39 (46.2%) vs 195/375 (52.0%). 'My' OR = 0.79 (95%-CI 0.41 to 1.53) [= (18/21)/(195/180)]. 'My' risk difference (or 'Difference in proportions') would be -0.06 (95%-CI -0.22 to +0.11) (or, if you wish, -6% (-22% to +11%), which is directly interpretable. Please, check all calculations (or, preferably) replace all ORs with RDs (+95%-CIs).

Apologies for this, although the maths was checked there was a mistake in the input that was not noted.

However, we have replaced ORs with differences of the mean and 95% CIs as requested.

3. I'm also very confused by the use of the paired t-test (page 7 - lines 6-7: A paired t-test was used to compare citation numbers and odds ratios used all other comparisons at 95% confidence intervals). A paired t-test is used for paired (or dependent) data, e.g. the difference between baseline and follow-up in a series of patients. In this study, we have independent data and, therefore, the independent t-test should be used. This also produces estimates of the difference in means (e.g. mean number of citations) and a 95%-CI. NB: this requires normally distributed data. Let's look at page 7 - lines 30-32: "Citation numbers were similar in emergency (mean 21.75, range 0-98) and elective (mean 21.59, range 0-306) surgical trials (difference 0.16 95%CI -16.7 to 16.9)". I can't replicate this (not sure if this is correct; the 95% CI is extremely wide). If you calculate a difference of means with a 95% CI, one assumes Normal distributions. In that case one should mention the standard deviations (instead of or additional to the ranges that are reported now). However, I expect that these data are severely non-normal, so a non-parametric statistical test is indicated (comparing medians instead of means) and medians and ranges should be reported. One could question, however, whether in this case a test is indicated (in view of the trivial difference of 0.16 citations).

This seems to be a sensible suggestion and our original manuscript had indeed left out a statistical test. We have removed it.

Specific remarks

4. Title: to me, this is not a systematic review. What about a (historical) cohort study (of surgical trials registered in trial registries)?

We agree with this remark and have edited as suggested.

5. Page 7 - Table 1: the totals (360-39) don't match with the text (375-39). Add a row with Other (or Remaining or Not classified) to match the totals in the text.

This has been changed.

6. Page 8 - Lines 41-42: "Unpublished emergency surgical trials were equally likely to be terminated early compared with elective trials (33.3% vs 15.6%; OR 2.4, 95%CI 0.88 - 6.40)." This is an incorrect conclusion (and an incorrect OR), which also applies to other instances, e.g. in "Unpublished trials in both groups were equally likely to still be planning publication (52.4% vs 70.1%; OR 0.50, 95%CI 0.20 - 1.25)". Not-significant is not similar to equality or lack of effect. Formally one should say "A difference of having been terminated early between emergency and elective trials could be demonstrated nor refuted."

Apologies for this mistake. These conclusions had been made on the outcomes of the chi squared test, which did indeed show no (significant) difference. Unfortunately, this was overlooked when calculating the (incorrect) ORs. The MDs and 95% CIs calculated are in line with these conclusions.

7. Page 8-9: Table 2 is not referred to in the text (or I missed it).

Apologies, we thought we had added this as the table changed, although it was somehow missed off.

8. Page 9 - Lines 23-24: "Of published trials, there was no significant differences in sponsorship (supplementary table 1)". Please, correct the English.

Thank you, we have changed to 'were'

9. Discussion: please, check the conclusions (see also my previous remarks about incorrect inferences). This also applies to the abstract.

Apologies again for this mistake. These conclusions had been made on the outcomes of the chi squared test, which did indeed show no (significant) difference. Unfortunately, this was overlooked when calculating the (incorrect) ORs. The MDs and 95% CIs now calculated are in line with these conclusions.

10. Figure 2: it's very confusing to have another format for C (which covers the whole width of the page). The three (or two) curves could be presented in one graph. In addition, no methods are mentioned for the Kaplan-Meier curves / time to publication. In the results, there's only one line ("The rate of publications, in each group and overall, are shown in figure 2." without any interpretation (or log-rank test to test the hypothesis that time to publication is equal between the two types of surgery).

We have changed to format of the graphs to be all on the same scale. However, the reasoning behind creating the Kaplan meier graphs from the previous comments was to show the plateauing of the rate of publication, not the difference between the two groups. Methods have been added to reflect this.

Overlooked in the previous version (apologies):

11. Please, replace "randomised control trials" with "randomised controlled trials" (overlooked in the previous version).

This has been done

12. Page Line 41: "Fewer randomised clinical trials": please, be consistent with labelling (- >"Fewer randomised controlled trials")

Thank you, this has been ammended.

13. Page 4 Line 25: "The secondary objectives were to compare rates of ...". You don't compare rates (which have an element of time in it: number of events / person time), but proportions. Please, correct.

Thank you, this has been ammended.

14. Page 5: two search strategies are presented. I may miss the point completely, but to me, strategy 2 completely covers strategy 1. Please, explain the difference (and the need for strategy 1).

As we searched two separate registries, each had their own search interface. In Clinicaltrials.gov, one can exclude certain things from the search, whereas in ISRCTN the search function was much more limited.

15. Page 6, line 11: The searches were categorised -> The identified RCTs were categorised
Thank you, this has been ammended.

16. Page 6 - Line 52-53: The second email was sent to all unpublished trials -> to the contact persons of unpublished trials
Thank you, this has been ammended.

17. Page 7 - Line 28: "The rate of publications, in each group and overall, are shown in". As mentioned before: rates are about incidence densities. Proportion is a better label. English grammar should be corrected ('rate are shown' -> proportionS are shown).

We have changed this as suggested.

18. Page 8 – Lines 23-25: "The majority of these publications were [was?] cited within trial registrations (n=110)." Do you mean: The majority of publications was found / identified / discovered by citations that were included in the trial registrations?

We have amended this to 'Most publications were identified from citations included within trial registrations.'

19. Page 8 – Lines 32-27: "Reasons for non-publication of emergency and elective trials respectively were and unknown". How can a reason be unknown? Didn't the trialists know why they stopped the trial or did they not answer this question (-> 'no reason provided')? They are listed as 'unknown' in the registry. Presumably from lack of data entry although it's not clear.

VERSION 3 – REVIEW

REVIEWER	Bob Siegerink CSB, charite,berlin
REVIEW RETURNED	03-May-2018

GENERAL COMMENTS	The authors have addressed all my comments. However, the KM that is presented seems not a true KM. KM curves only have horizontal and vertical lines, as survival is a binary concept. A new version of the graphs needs to be submitted.
---

REVIEWER	Rob Scholten Julius Center for Health Sciences and Primary Care University Medical Center Utrecht, The Netherlands
REVIEW RETURNED	08-May-2018

GENERAL COMMENTS	Congratulations: very much improved. Very readable! Just a few minor (cosmetic) point, which I trust the authors will deal with.  1. Most analyses have non-significant results. Non-significant is not similar to 'no difference' or 'similar'. E.g. in the abstract "The proportion of trials successfully published were similar between emergency and elective surgery (0.462 vs 0.520, mean difference - 0.0584 95%CI -0.237-0.120)" the difference could be 23.7% less or 12% more, which is far from 'similar'. You may wish to reword into "The proportion of trials successfully published between emergency and elective surgery did not differ significantly" or "A difference in the proportion of trials successfully published between emergency and elective surgery could be demonstrated nor refuted". If changed, change such wording throughout in the text. 2. I'm not sure about the policy of the Journal, but three decimals (as in "0.462 vs 0.520, mean difference -0.0584 95%CI -0.237-0.120") is somewhat overdone. Use two decimals of express the results using percentages (-> 46.2% vs 52.0% - etc.) 3. You may wish to add a heading "Surgery trials" (at the pertinent spot in the first section of the Results) and "Adjunct trials" on page 9, line 53. 4. Page 7, lines 34-36: you may wish to add the unit/scale: months? 5. Page 10, line 21-22: delete "remained" in "but found that 48% of all surgical trials remained published at 38 months." Or should it be "but found that 48% of all surgical trials remained UNpublished at 38 months."?
---

	6. Legend of Figure 2 “Kaplan-Meier curves showing publication over time of emergency (blue) and elective (green) surgical trials. Only trials published on or after the July 2012 were included, there is a cohort of trials that are registered after they started/published.”. Not sure what is meant with “, there is a cohort of trials that are registered after they started/published.”
--	--

VERSION 3 – AUTHOR RESPONSE

Reviewer: 2

Reviewer Name: bob siegerink

Institution and Country: CSB, charite,berlin

Please state any competing interests or state ‘None declared’: none declared

Please leave your comments for the authors below

The authors have addressed all my comments. However, the KM that is presented seems not a true KM. KM curves only have horizontal and vertical lines, as survival is a binary concept. a new version of the graphs needs to be submitted.

Thank you for pointing this out, they have been edited accordingly.

Reviewer: 1

Reviewer Name: Rob Scholten

Institution and Country: Julius Center for Health Sciences and Primary Care | University Medical Center Utrecht, The Netherlands

Please state any competing interests or state ‘None declared’: None declared.

Please leave your comments for the authors below

Congratulations: very much improved. Very readable!

Thank you!

Just a few minor (cosmetic) point, which I trust the authors will deal with.

1. Most analyses have non-significant results. Non-significant is not similar to ‘no difference’ or ‘similar’. E.g. in the abstract “The proportion of trials successfully published were similar between emergency and elective surgery (0.462 vs 0.520, mean difference -0.0584 95%CI -0.237-0.120)” the difference could be 23.7% less or 12% more, which is far from ‘similar’. You may wish to reword into “The proportion of trials successfully published between emergency and elective surgery did not differ significantly” or “A difference in the proportion of trials successfully published between emergency and elective surgery could be demonstrated nor refuted”. If changed, change such wording throughout in the text.

We have clarified with the word ‘statistically’ where appropriate.

2. I’m not sure about the policy of the Journal, but three decimals (as in “0.462 vs 0.520, mean

difference -0.0584 95%CI -0.237-0.120”) is somewhat overdone. Use two decimals of express the results using percentages (-> 46.2% vs 52.0% - etc.)

This has been changed to two decimals. We felt expressing as percentages somewhat overwhelmed the results with % symbols.

3. You may wish to add a heading “Surgery trials“ (at the pertinent spot in the first section of the Results) and “Adjunct trials” on page 9, line 53.

These have been added.

4. Page 7, lines 34-36: you may wish to add the unit/scale: months?

The unit is number of citations, we have attempted to clarify this by changing ‘citation numbers’ to ‘number of citations.’

5. Page 10, line 21-22: delete “remained” in “but found that 48% of all surgical trials remained published at 38 months.” Or should it be “but found that 48% of all surgical trials remained UNpublished at 38 months.”?

Thank you for picking that up, it should be UNpublished and this has been amended.

6. Legend of Figure 2 “Kaplan-Meier curves showing publication over time of emergency (blue) and elective (green) surgical trials. Only trials published on or after the July 2012 were included, there is a cohort of trials that are registered after they started/published.”. Not sure what is meant with “, there is a cohort of trials that are registered after they started/published.”

At the beginning of the registration period we searched some papers had already been published, thus some must have been retrospectively registered. Since this can be inferred from the graph itself, we have removed this clause to avoid confusion.